# In-cell NMR suggests that DNA i-motif levels are strongly depleted in living human cells

Pavlína Víšková[1,2,10], Eva Ištvánková [1,2,10], Jan Ryneš[1], Šimon Džatko[1,3], Tomáš Loja [1], Martina Lenarčič Živković [1,4], Riccardo Rigo[1,5], Roberto El-Khoury [6], Israel Serrano-Chacón[7], Masad J. Damha [6], Carlos González [7], Jean-Louis Mergny[8,9], Silvie Foldynová-Trantírková [1,8] ✉ & Lukáš Trantírek [1] ✉

I-Motifs (iM) are non-canonical DNA structures potentially forming in the accessible, single-stranded, cytosine-rich genomic regions with regulatory roles. Chromatin, protein interactions, and intracellular properties seem to govern iM formation at sites with i-motif formation propensity (iMFPS) in human cells, yet their specific contributions remain unclear. Using in-cell NMR with oligonucleotide iMFPS models, we monitor iM-associated structural equilibria in asynchronous and cell cycle-synchronized HeLa cells at 37 °C. Our findings show that iMFPS displaying $pH_T < 7$ under reference in vitro conditions occur predominantly in unfolded states in cells, while those with $pH_T > 7$ appear as a mix of folded and unfolded states depending on the cell cycle phase. Comparing these results with previous data obtained using an iM-specific antibody (iMab) reveals that cell cycle-dependent iM formation has a dual origin, and iM formation concerns only a tiny fraction (possibly 1%) of genomic sites with iM formation propensity. We propose a comprehensive model aligning observations from iMab and in-cell NMR and enabling the identification of iMFPS capable of adopting iM structures under physiological conditions in living human cells. Our results suggest that many iMFPS may have biological roles linked to their unfolded states.

DNA has a well-known propensity to adopt alternative non-B form conformations, including the intercalated motif (i-motif, iM) structure[1]. This quadruple-helical structure forms within C-rich DNA and is stabilized by intercalating hemi-protonated C:CH+ base pairs[1,2] (Fig. 1A, B). Sequences with iM-forming potential (iMFPS) are frequently found within evolutionarily conserved human genome regions associated with regulatory functions[3]. Biophysical analyses of iMFPS

derived from (neo-)centromeres, telomeres, and promoter regions of (onco-)genes and synthetic constructs demonstrated that the stabilities of intramolecular iMs depend on the number of cytosines in the i-motif core, the length and composition of loops, chemical modifications, and a broad range of environmental factors, particularly molecular crowding, pH, and temperature[4–11]. The dependency of the iMs' stability on the abovementioned factors has been employed

[1]Central European Institute of Technology, Masaryk University, 625 00 Brno, Czech Republic. [2]National Centre for Biomolecular Research, Masaryk University, 625 00 Brno, Czech Republic. [3]Centre for Advanced Materials Application, Slovak Academy of Sciences, 845 11 Bratislava, Slovakia. [4]Slovenian NMR Centre, National Institute of Chemistry, SI-1000 Ljubljana, Slovenia. [5]Pharmaceutical and Pharmacological Sciences Department, University of Padova, 35131 Padova, Italy. [6]Department of Chemistry, McGill University, Montreal, QC H3A0B8, Canada. [7]Instituto de Química Física 'Blas Cabrera', CSIC, C/Serrano 119, 28006 Madrid, Spain. [8]Institute of Biophysics, Czech Academy of Sciences, Brno 612 00, Czech Republic. [9]Laboratoire d'Optique & Biosciences, Institut Polytechnique de Paris, Inserm, CNRS, Ecole Polytechnique, Palaiseau 91120, France. [10]These authors contributed equally: Pavlína Víšková, Eva Ištvánková. ✉e-mail: silvie.trantirkova@ceitec.muni.cz; lukas.trantirek@ceitec.muni.cz

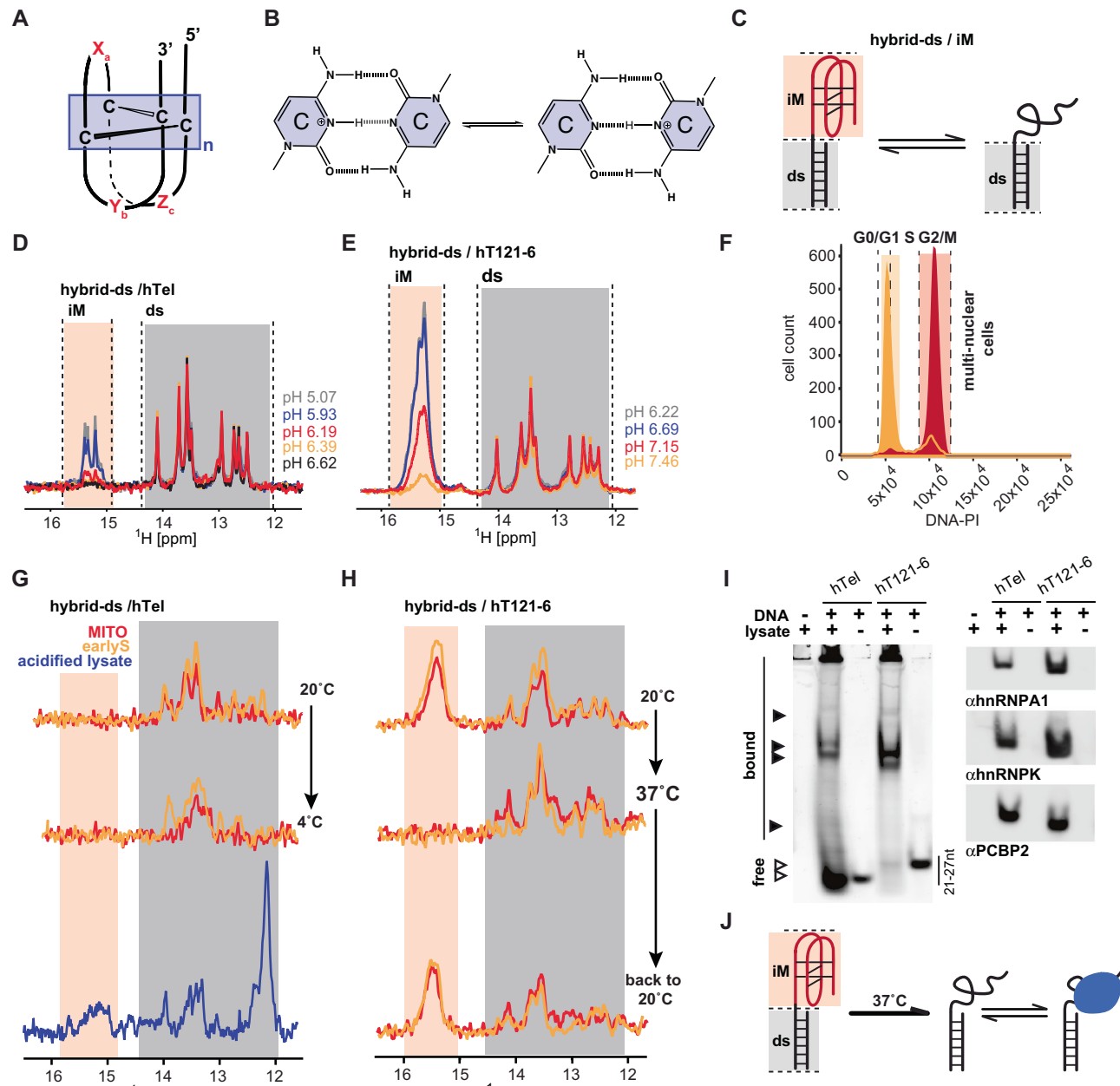

**Fig. 1 | Hybrid-ds/hTel and hybrid-ds/hT121-6 do not form iM in cells at 37 °C.**
**A** Schematic representations of the DNA i-motif structure with the minimal i-motif segment highlighted in the box. $X_n$, $Y_n$, and $Z_n$ mark i-motif loop regions of variable length. **B** C:CH[+] base-pair, according to ref. 2. **C** Hybrid-ds/iM reporter of iM-associated structural equilibrium; ds and iM correspond to double-stranded and i-motif segments, respectively. **D**, **E** show overlays of the imino regions of 1D $^1$H NMR spectra of hybrid-ds/hTel and hybrid-ds/hT121-6, respectively, acquired at 20 °C as a function of the pH in-vitro (IC buffer: 25 mM $KPO_i$, 10.5 mM NaCl, 110 mM KCl, 1 mM $MgCl_2$, 130 nM $CaCl_2$). Spectral regions specific for the imino protons involved in C:CH[+] (iM) and Watson-Crick (ds) base pairs are indicated in light orange and gray, respectively. **F** shows propidium iodide (PI) DNA content staining of cells transfected with (FAM-)hybrid-ds/hT121-6 and synchronized in M (red) and early-S (green) cell-cycle phase. **G**, **H** show imino regions of 1D $^1$H in-cell NMR

spectra of hybrid-ds/hTel and hybrid/hT121-6, respectively, acquired as a function of the temperature (indicated) in mitotic- (red, MITO) and early-S (green, earlyS) synchronized cells. The displayed in-cell NMR spectra are representative images of two independent experiments. In blue is the NMR spectrum of the hybrid-ds/hTel acquired in the acidified (pH < 6) cell lysate prepared from the respective in-cell NMR sample at 4 °C (folding control). **I** LEFT: native PAGE of the Cy3-labeled hTel and hT121-6 in the absence and the presence of lysates from HeLa cells visualized via the Cy3-fluorescence. RIGHT: immuno-stained electroblots (PVDF membrane) of the native PAGE with αhnRNP A1, αhnRNP K, and αPCBP2. The gel and blot are representative images of three independent experiments. Source data are provided as a Source Data file. **J** Projected representation of structural equilibria involving hybrid-ds/iMs in living cells.

in designing biosensors for applications in biotechnology and nanotechnology[12,13]. The iMs are relatively stable at acidic pH, low ionic strength, and dehydrating conditions. However, the propensity for iMs is compromised under expected physiological conditions comprising relatively high ionic strength (~150 mM), physiological temperatures (~37 °C for human cells), and slightly alkaline pH (intracellular pH in

human cells ~7.0–7.4)[4], thereby raising concerns and controversy about the biological relevance of iMs.

Conversely, the potential presence of iM within living organisms, including humans, has been supported by several lines of evidence, including the discovery of iMFPS binding proteins[14–17]; thermodynamic stabilization of iM by epigenetic modifications in vitro[5,18,19] and under

**Table 1 | List of iM forming sequences used in this study**

| iM | Sequence (5′- > 3′) | iM / hybrid-ds/iM | |
|---|---|---|---|
| | | $T_m^{in-vitro}$ (pH 7) | $pH_T^{in-vitro}$ (20 °C) |
| hTel | d($C_3$TAA)$_3C_3$ | n.d./<br>n.d. | 6.20 ± 0.02/<br>6.09 ± 0.01 |
| hT121-6 | d($C_6$T)$_3C_6$ | 24.9 ± 0.5/<br>28.0 ± 0.4 | 6.94 ± 0.03/<br>6.81 ± 0.02 |
| hBcl-2 | d(CAGC$_4$GCTC$_3$GC$_5$T$_2$C$_2$TC$_3$GCGC$_3$GC$_4$T) | 12.9 ± 0.5/<br>- | 6.48 ± 0.03<br>- |
| hPDGFa | d(C$_2$GCGC$_4$TC$_5$GC$_5$GC$_5$GC$_{13}$) | 28.8 ± 1.2/<br>27.6 ± 0.9 | 6.96 ± 0.03/<br>6.96 ± 0.05 |
| hRAD17 | d(GCT$_2$CTAGTCA$_2$TC$_2$AC$_9$GC$_9$G$_2$A) | 33.8 ± 0.5/<br>34.9 ± 0.6 | 7.08 ± 0.02/<br>7.10 ± 0.03 |

In the hybrid-ds/iM constructs, the ds segment d(GCTTCTAGTCAAT*).d(TTGACTAGAAGC) was attached to the 5′-end of iM. T* corresponds to the unpaired spacer nucleotide between ds and iM segments. For the complete list of sequences and modifications, see Supplementary Table 1. $T_m^{in-vitro}$ and $pH_T^{in-vitro}$ corresponds to melting temperature and transitional pH measured in-vitro (25 mM KPO$_i$, 10.5 mM NaCl, 110 mM KCl, 1 mM MgCl$_2$, 130 nM CaCl$_2$) at pH = 7 and T = 20 °C, respectively. The $T_m^{in-vitro}$ and $pH_T^{in-vitro}$ values for iM and corresponding hybrid-ds/iM construct are typeset on top and bottom, respectively. Error bars for the $pH_T^{in-vitro}$ values represent the 95% confidence interval of the calculated fit. Errors reported for $T_m^{in-vitro}$ values correspond to the standard error of the mean for each triplicate experiment. The experimental data are displayed in Supplementary Figs. 1 and 2.

conditions imitating a crowded cellular environment[8,20,21]; and the promotion of iM formation from double-stranded DNA under conditions emulating mechanistic forces connected with DNA helix unwinding and nanoconfinement in the chromatin context[22–25]. Several indirect findings also suggested that iM formation may play a role in regulating replication and transcription[15,26,27].

Undoubtedly, the recent development and applications of an iM-specific antibody (iMab) have provided breakthrough contributions regarding iM biological relevance using iMab-detected iMs in plant[28] and human genomic DNA[29–31]. Most recently, Richter's group employed iMab and revealed that iMs are enriched in actively transcribing gene promoters and open chromatin regions, and they overlap with R-loops[32]; all these findings supported that iMs are associated with transcriptional regulation in live human cells. However, the detection method employed by iMab relies on a combination of cell membrane permeabilization, which may disturb environmental conditions in the intracellular space, and prolonged times of iMab staining at a close-to-zero temperature[29–32], which is a decisive iM formation promoting factor[3,13,33]. The extent to which iMab-detected iMs in chromatin of permeabilized cells at close-to-zero temperature form iM structures in the intracellular space of living human cells at physiological temperature has remained unknown.

In this work, we assess how intracellular environmental factors and physiological temperatures affect the formation of intramolecular structures (iMs) within living cells. Employing an in-cell NMR-based technique, we examine the dynamic equilibrium of iMs in various oligonucleotide-based iMFPS models within asynchronous, M-phase, and early-S-phase-synchronized human (HeLa) cells, all maintained at 37 °C. Our in-cell NMR data reveal that certain iMs can indeed form within living human cells at normal body temperature, and their prevalence is influenced by the cell cycle-dependent properties of the intracellular environment; our investigation uncovers a dual origin for the cell cycle dependency of iM formation in cells. Simultaneously, we demonstrate that iM formation occurs only at a minute fraction of genomic sites displaying a propensity for iM formation. Finally, we present a comprehensive model for iM formation in genomic iMFPS within human cells, consolidating findings from iMab and in-cell NMR techniques. This model allows for the identification of specific sites in the human genome that have the potential to adopt iM structures under physiological conditions within living human cells.

## Results

To achieve in-cell NMR readout in a defined cell-cycle phase, we adapted the original in-cell NMR method[34] to introduce a model/ reporter iMFPS oligonucleotide(s) into cells synchronized at cell cycle specific phases, M- and early-S. To enable a quantitative readout of iM-associated structural equilibria, we used hybrid iM-based constructs (hybrid-ds/iM). In these constructs, iM (reporter) elements are interlinked with a double-stranded (ds) DNA fragment (Fig. 1C). In comparison (relatively) to the iM elements, the ds module demonstrates insensitivity to moderate environmental changes (detailed below) and serves as an internal NMR reference to normalize the i-motif/single-strand equilibrium within cells, independent of cell transfection efficiencies.

The hybrid-ds/hTel and hybrid-ds/hT121-6 constructs were employed in the initial phase of the study (Table 1, Supplementary Table 1). The in-vitro-validated iM-modules of hTel and hT121-6 correspond to natively occurring sequences from the human genome[33,35]. They represent an often-studied class of iMFPS characterized by four equivalent C-tracts separated by a spacer sequence of constant length and nucleotide composition. While hTel corresponds to iMFPS previously identified using iMab[29], the hT121-6 is one of the most stable (pH-resistant) iMs from this class characterized in-vitro to date[33].

The capacity of the hybrid-ds/iM constructs to report on i-motif/ unfolded single-strand equilibrium was first confirmed in-vitro (IC buffer; 25 mM KPO$_i$, 10.5 mM NaCl, 110 mM KCl, 1 mM MgCl$_2$, 130 nM CaCl$_2$); compared to hTel and hT121-6, their hybrid analogs (hybrid-ds/ iM) showed a slightly decreased pH resistance at 20 °C ($\Delta pH_T^{in-vitro}$ ~0.1) (Table 1). The hybrid-ds/hT121-6 showed increased thermodynamic stability compared to hT121-6 ($\Delta T_m^{in-vitro}$ ~3 °C) (Table 1). Note: $T_m^{in-vitro}$ could not be determined for hTel and hybrid-ds/hTel, as they do not form iM at pH 7 (Supplementary Fig. 1). Nevertheless, in agreement with the previous report[36], we confirmed the capacity of hybrid-ds/hTel to form iM at 4 °C and neutral pH; we observed the appearance of the iM-specific signals confirming iM formation in hybrid-ds/hTel at time >4 h (Supplementary Fig. 3). Figure 1D, E shows the in-vitro hybrid-ds/hTel and hybrid-ds/hT121-6 NMR spectra recorded as a function of increasing pH, an environmental parameter known to modulate iM stability[4]. In response to increasing pH, the NMR signals intensity between 15–16 ppm, corresponding to the imino protons involved in C:CH$^+$ base-pairs, decreased, while the intensity of NMR signals corresponding to the imino protons involved in Watson-Crick (WC) pairs (~12–14.5 ppm) remained essentially the same. The trend observed in the in-vitro NMR spectra (Fig. 1D, E) was reversible, confirming that the iM element could fold/unfold in the presence of the ds-segment and that signals from this segment could be exploited as an internal (normalization) reference. The pH-resolved CD experiments conducted in parallel corroborated this interpretation (Supplementary Fig. 4). Additionally, temperature-resolved NMR spectra obtained in IC buffer (Supplementary Fig. 5A) and crude cellular homogenate

(Supplementary Fig. 5B) confirmed the ds-segment's minimal sensitivity to temperature changes in a range between 4 to 37 °C, in contrast to the iM segment (hT121-6), in both in vitro and cell-like settings. These data further supported the ds-segment's suitability to normalize the i-motif/single-strand equilibrium within cells.

Next, the hybrid-ds/hTel and hybrid-ds/hT121-6 were introduced separately into an asynchronous suspension of HeLa cells, double-thymidine-treated cells (arrested at the early-S phase of the cell cycle), and nocodazole-treated cells (arrested at M-phase of the cell cycle). The transfection of the hybrid-ds/iMs had a small-to-moderate impact on cell viability/membrane integrity (Supplementary Figs. 6A and 7A). The introduced hybrid-ds/iMs localized almost quantitatively in the cell nucleus (early-S-phase cells) and were uniformly dispersed through the intracellular space of M-phase arrested cells, as expected for cells lacking the nuclear membrane (Supplementary Figs. 6B and 7B). The propidium iodide (PI) and MPM2-Cy5 staining quantitatively confirmed the synchronization efficiency (Fig. 1F, Supplementary Figs. 6C, D and 7C, D).

The spectral patterns of the imino regions of ds/hTel and hybrid-ds/hT121-6 1D $^1$H in-cell NMR spectra acquired in asyn-, early-S and M-cells at 20 °C showed imino signals from the ds segment (12.0–14.2 ppm), confirming the hybrid-ds/iMs presence in cells (Fig. 1G, H, and Supplementary Fig. 6E). In line with expectations from in-vitro data, the hybrid-ds/hTel in-cell NMR spectrum was devoid of iM-specific signal suggesting hTel segment is unfolded in cells at this temperature (Fig. 1G, Supplementary Fig. 6E). As expected from in-vitro data at a short incubation time (about 10 min), the iM-specific signal in the in-cell NMR spectra were not observed even upon reducing the temperature to 4 °C (Fig. 1G). The presence of iM-specific signals in the NMR spectrum of the acidified lysate prepared from in-cell NMR sample corroborated that the absence of iM signal(s) in hybrid-ds/hTel in-cell NMR spectra acquired at 4 and 20 °C reflects the construct's unfolding, not its degradation (Fig. 1G, Supplementary Fig. 6E).

In contrast, the hybrid-ds/hT121-6 in-cell NMR spectra recorded at 20 °C displayed overlapping signals at ~15.4 ppm from C:CH$^+$ base pairs, indicating iM formation in early-S and M-synchronized cells (Fig. 1H). Importantly, no iM signals were detected, in contrast to the signals from the ds-reference-segment, in the hybrid-ds/hT121-6 in-cell NMR spectra acquired at 37 °C (Fig. 1H), suggesting the iM segment unfolding. Upon reducing the temperature to 20 °C, the initial iM signal intensities in the hybrid-ds/hT121-6 in-cell NMR spectra were re-established, evidencing that the lack of iM signals in the in-cell NMR spectrum recorded at 37 °C was due to the construct unfolding (Fig. 1H). Noteworthy, the corresponding in-cell NMR spectra measured in non-synchronized (asyn-) cells revealed identical phenotype to that observed in early-S and M-phase synchronized cells regardless of the method used for in-cell NMR sample handling (pelleted cells vs. bioreactor) and acquisition time (10 vs. 30 min) (Supplementary Fig. 7F vs. Supplementary Fig. 8). Note: for a discussion of potential technical issues that may impact the interpretation of the in-cell NMR data, please refer to Supplementary Table 2 and recent literature[37,38].

Matching signal patterns (signal positions)[39] and similar linewidths[40] among the corresponding signals in in-vitro and in-cell NMR spectra acquired at 20 °C (Fig. 1D, E, G, H) evidence that in-cell NMR readout reports on the intracellular space's free (unbound) hybrid-ds/iMs. However, agarose EMSA performed on the lysate prepared from asynchronous HeLa cells transfected with the iMs revealed the existence of two distinct iMs fractions in cells: one corresponding to the free (unbound) iMs and one possibly corresponding to the constructs embedded in complexes with endogenous proteins (Supplementary Fig. 9A). To confirm the protein binding to our constructs, we performed the shift-western blot[41] (WB) analysis of PAGE-resolved lysate prepared from HeLa cells transfected with hTel and hT121-6 (Supplementary Fig. 9B, C), using antibodies against three promiscuous and previously described DNA C-rich binding

proteins[14,16,17,42], namely hnRNP K, hnRNP A1, and PCBP2. The WB confirmed the binding of all three proteins to hTel and hT121-6 (Fig. 1I, Supplementary Fig. 9C).

To extend these observations, we probed iM structural equilibria in asynchronous cells for three in-vitro-validated and previously studied iMFPS from the promoter regions of hBcl-2[27], hPDGFa[43], and hRAD17[44] genes (Table 1). These constructs were chosen for the following reasons: i] these iMFPS notably differ in stability/pH resistance (cf. Table 1, Supplementary Figs. 1 and 2); their $T_m/pH_T$ values cover a range typically displayed by often-studied iMFPS, and ii] hBcl-2, hPDGFa, and hRAD17 represent iMFPS with distinctive sequence/ structure properties: hBcl-2 iM co-exists in equilibrium with an alternative structure (hairpin)[27], hRAD17 represents an intriguing class of poly-C iMFPS showing exceptional pH resistance[9], and hPDGFa[43] sits on the borderline between C-tract-spacer repeats and poly-C based iMFPS class (Table 1).

For the in-cell NMR study, analogously to hTel and hT121-6, the ds-segment was attached to hPDGFa and hRAD17 iMFPS. A comparison of the NMR/CD spectra of hPDGFa and hRAD17 and their hybrid counterparts (hybrid-ds/hPDGFa and hybrid-ds/hRAD17) confirmed that attachment of the ds segment did not compromise the folding into iM (Fig. 2A, Supplementary Fig. 4) and had a minimal-to-negligible impact on the respective $pH_T^{in-vitro}$ and $T_m^{in-vitro}$ values (Table 1, Supplementary Figs. 1 and 2). Conversely, hBcl-2 could be analyzed without the ds segment addition: in this case, the position of the iM-associated equilibrium can be assessed based on the relative signal ratio between hairpin (at ~13 ppm) and iM-specific NMR signal (at ~15.5 ppm) (Fig. 2A).

Next, we introduced the hBcl-2, hybrid-ds/hPDGFa, and hybrid-ds/ hRAD17 constructs in the asynchronous HeLa cells. The introduction did not significantly compromise cells' viability/membrane integrity (Supplementary Figs. 10A and 11A). The introduced constructs predominantly localized to the cell nucleus (Supplementary Figs. 10B and 11B). At 20 °C, the in-cell NMR spectra of hybrid-ds/PDGFa and hybrid-ds/RAD17 displayed, next to the ds-segment signals, a composite signal at ~15.5 ppm (Fig. 2B), indicating iM formation. In contrast, only the hairpin-specific signals (at ~13 ppm) were observed in the hBcl-2 in-cell NMR spectrum, suggesting that iM-hairpin equilibrium is shifted to a hairpin state, in agreement with in-vitro data acquired at close-to-neutral pH (Fig. 2A vs. 2B).

The behavior of hBcl-2, hybrid-ds/PDGFa, and hybrid-ds/RAD17 in cells at 37 °C differed considerably: neither iM- nor hairpin-specific signals were observed in the hBcl-2 in-cell NMR spectrum (Fig. 2B), suggesting that hBcl-2 is completely unfolded or completely /partially degraded. Observing restoration of iM and hairpin signals in the in-cell NMR spectrum acquired of hBcl-2 at 4 °C (Fig. 2B) confirmed that the spectral phenotype at 37 °C resulted from the construct unfolding. Hybrid-ds/PDGFa and hybrid-ds/RAD17 in-cell NMR spectra acquired at 37 °C showed characteristic signals from the ds-reference-segment (Fig. 2B), confirming the constructs' presence in cells. However, no iM signals were detected in the hybrid-ds/PDGFa in-cell NMR spectrum (Fig. 2B), suggesting the iM segment unfolding or degradation. Conversely, the hybrid-ds/RAD17 in-cell NMR spectrum displayed a signal at ~15.5 ppm, indicating iM formation at 37 °C (Fig. 2B). Upon reducing the temperature to 20 °C, the initial iM signal intensity in the hybrid-ds/PDGFa in-cell NMR spectrum was reestablished (Fig. 2B), evidencing that iM signal absence at 37 °C resulted from unfolding.

Similarly to the hTel and hT121-6, EMSA in agarose gel (Supplementary Fig. 9A) and the western blot analysis of PAGE resolved lysate prepared from cells transfected with Cy3- hBcl-2, hPDGFa, and hRAD17 (Supplementary Fig. 9B) confirmed the binding of these constructs to endogenous hnRNP K, hnRNP A1, and PCBP2 proteins (Fig. 2C, Supplementary Fig. 9C).

Observing the iM signals in the hybrid-ds/hRAD17 in-cell NMR spectrum at 37 °C allowed us to assess the influence of the intracellular environmental factors specific to the M- and early-S cell cycle phases

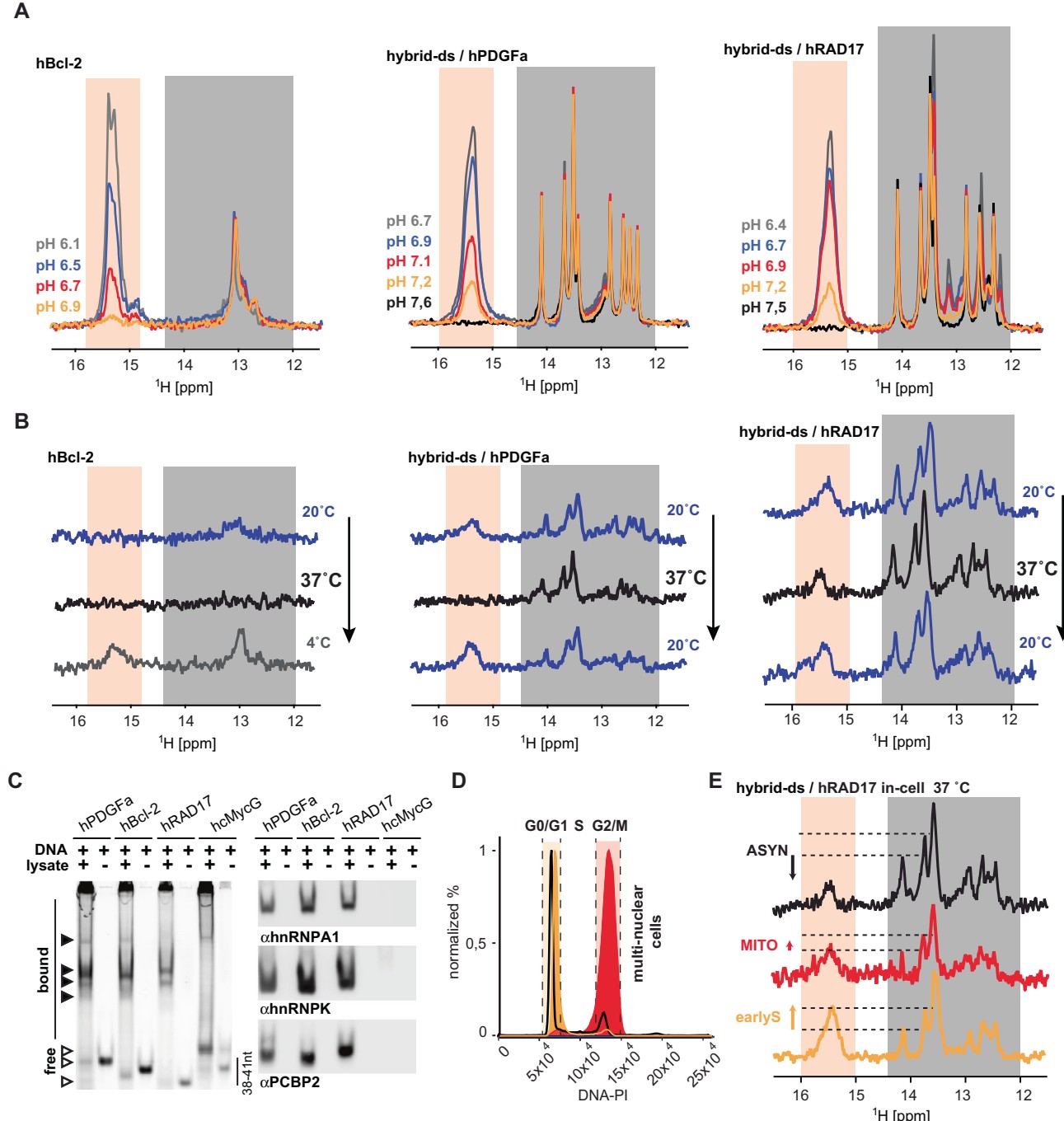

**Fig. 2 | Hybrid-ds/hRAD17 folds into iM in cells; its iM levels change depending on the cell cycle phase. A** shows overlays of the imino regions of 1D $^1$H NMR spectra of hBcl-2, hybrid-ds/hPDGFa, and hybrid-ds/hRAD17 acquired at 20 °C as a function of the pH in-vitro (IC buffer: 25 mM KPO$_i$, 10.5 mM NaCl, 110 mM KCl, 1 mM MgCl$_2$, 130 nM CaCl$_2$). Note: the Watson-Crick part of the NMR spectra of hybrid-ds/RAD17 display some extra signals compared to the other hybrid-ds/iM constructs. The extra signals come from GC base pairs in the hRAD17 iM segment. **B** shows imino regions of 1D $^1$H in-cell NMR spectra of hBcl-2, hybrid-ds/hPDGFa, and hybrid-ds/hRAD17 acquired as a function of the temperature (indicated) in asynchronous HeLa cells. **C** LEFT: native PAGE of the Cy3-labeled hBcl-2, hPDGFa, hRAD17, and G-quadruplex forming control (hcMycG) in the absence and the presence of lysates

from HeLa cells visualized via the Cy3-fluorescence. RIGHT: immuno-stained electroblots (PVDF membrane) of the native PAGE with αhnRPNP A1, αhnRNP K, and αPCBP2. The gel and blot are representative images of three independent experiments. Source data are provided as a Source Data file. **D** shows PI DNA content staining of cells transfected with (FAM-)hybrid-ds/hRAD17: asynchronous (black line) and synchronized in M (red, G2/M) and early-S (green, G0/S1) cell-cycle phase. **E** shows imino regions of 1D $^1$H in-cell NMR spectra of hybrid-ds/hRAD17 acquired at 37 °C in HeLa cells: asynchronous (black) and synchronized in M (red, MITO) and early-S (green, earlyS) cell-cycle phase. The dashed-horizontal lines highlight iM level differences among early-S phase, M-phase, and asynchronous cells. The displayed in-cell NMR spectra are representative images of two independent experiments.

on the iM formation. The hybrid-ds/hRAD17 was introduced separately into cells arrested at the early-S and M-phase of the cell cycle (Fig. 2D, E, Supplementary Fig. 10). The hybrid-ds/hRAD17 in-cell NMR spectra recorded at 37 °C in early-S and M-synchronized cells displayed characteristic iM signals. Notably, comparing the iM signal intensities relative to the ds-segment reference in the hybrid-ds/hRAD17 in-cell NMR spectra revealed a more pronounced formation of iM in the early-S cell cycle-synchronized cells compared to those in asynchronous (most of the cells in $G_0/G_1$) and M-phase-synchronized cells (Fig. 2E), suggesting that the cell cycle-specific intracellular environment contributes to the iM levels modulation in cells.

## Discussion

iM formation, accounting for the transition from the disordered (coil) to ordered (iM) structure in the context of strand-separated chromatin regions in living human cells at physiological temperature, is a subject of ongoing debate. Using an adapted in-cell NMR approach, we analyzed conformation equilibria for five oligonucleotide-based models of naturally occurring iMFPS, which differed by their $pH_T^{in-vitro}$ and $T_m^{in-vitro}$, in living asynchronous HeLa cells and those synchronized in M- and early-S phase of the cell cycle at 37 °C. We showed that the constructs that displayed $pH_T < 7$ under reference in vitro conditions did not form iM structures in cells at 37 °C. Using the hRAD17 model as a paradigm, we showed that iMFPS displaying $pH_T^{in-vitro} > 7$ occur in cells as a mix of folded (iM) and unfolded states; we showed that the populations of folded (iM) and unfolded states differed between the M and early S phases of the cell cycle. Consistent with the existing literature[14,17,42,45], we observed that our iMFSP models introduced into cells interacted with endogenous proteins.

Our observations qualitatively align with the results found in previous in-cell NMR studies, confirming the reversibility of iM folding/unfolding in cells[34] and the correlation between iM formation propensity in cells and iM in vitro stability[33]. In contrast to the later study, our experimental design allowed us to express the relationship between iM stability in vitro and iM formation in cells quantitatively. Our data confirm that some iM may form in cells, and the process is cell cycle-dependent, as found in studies using iMab[29,30].

However, major differences in these seemingly coherent observations were found when iMab and in-cell NMR data were investigated in greater detail. Cell cycle dependency, as observed by iMab, refers to changes in the number and localization of iMab-detected foci in the genome[29,30]; however, the in-cell NMR data revealed a shift in the conformation equilibrium in specific iMFPS (Fig. 2E). iMab-based observations of iM formation cell cycle dependency may be related to chromatin remodeling, which could correspond with cell cycle-dependent changes in the number and locations of iMFPS in the accessible, single-stranded (protein unbound) chromatin regions[46,47]. However, it's important to note that this interpretation is confined to the original iMab protocol that involves a cross-linking step[29]. Employing the modified protocol introduced by ref. 32, which excludes the cross-linking, iMab can also identify iMFPS in the single-stranded chromatin regions bound by proteins; iMab might outcompete cell-cycle specific proteins that bind to these iMFPS. By contrast, the in-cell NMR-based observations of iM formation cell cycle dependency are related to environmental oscillations, which could correspond with cell cycle-dependent changes in intracellular pH[48] or molecular crowding from macromolecules[49], for example.

Notably, over 600,000 iM-forming sites were detected in the human genome using iMab[31], including iMs that are clearly unstable in vitro at close-to-physiological pH (such as telomeric iM) (Supplementary Fig. 1)[13]. These detected sites comprise the majority (over 50%) of iMFSP expected in the human genome; however, our in-cell NMR data demonstrated that iM formation in cells at a physiological temperature only involves those with exceptional pH resistance, which

can be estimated, based on available data[50], to comprise less than 1% (~3000) of genomic iMFPS.

There are three fundamentally distinct explanations for this striking discrepancy between in-cell NMR and iMab-based observations. Firstly, this disparity may arise from a fundamental limitation of the in-cell NMR approach. The in-cell NMR data were obtained using oligonucleotide-based models of iMFPS. The stability of iMs formed from oligonucleotide iMFPS models might be lower than that of iMFPS embedded in native single-stranded chromatin. In native chromatin, mechanical forces act on the termini of iMFPS, and localized nano-confinement could contribute to iM stabilization. In simpler terms, the chromatin context might alter the energetics of iM formation by stabilizing the folded (iM) state. The potential extent of this stabilization can be inferred by comparing iMab-based observations of telomeric iM[29] with our in vitro/in-cell NMR data for the telomeric iMFPS model (hTel). In-cell NMR data showed no detectable levels of telomeric iM in cells (Fig. 1G). This observation is consistent with data acquired under reference in vitro conditions, indicating that the melting temperature ($T_m$) of hTel is well below 10 °C (Supplementary Fig. 1), confirming previous findings[13]. If the iMab-based experimental observation of telomeric iM[29] corresponds to its native formation at physiological temperature (37 °C), the chromatin-provided stabilization would need to account for a $T_m$ (in-vitro) equivalent to at least 25–30 °C (cf. Supplementary Fig. 1)[13]. Chromatin stabilization could potentially affect not only telomeric iMFPS but also all other iM-forming sites in the accessible, single-stranded chromatin regions. This increased iM stability would also manifest as a significant improvement in iM pH resistance, equivalent to approximately 1.5–2 pH units[33]. Consequently, iM formation would occur in a genomic context for most genomic iMFPS, in line with iMab-based data.

This explanation, which proposes a principal role of chromatin in thermodynamic and/or kinetic stabilizing noncanonical DNA structures in the context of accessible single-stranded DNA in cells, is supported by observations of promoting iM formation under nano-confinement, one of the potential features in chromatin. Using a single molecule, laser tweezer-based approach for a DNA construct based on telomeric DNA repeats, ref. 51. demonstrated that the confined space of the DNA origami nanocages causes a shift in equilibrium between unfolded and folded (iM) states; the shift depended on the nanocavity size. They showed that water molecules absorbed from the unfolded to the transition states are much fewer than those lost from the transition to the folded states and that the overall loss of water drives the folding of the iM in nanocages with reduced water activities[51]. This explanation aligns with previous observations of promoted iM formation in nanocavities from reversed micelles[25]. However, these effects on iM formation are likely not general phenomena and may depend on sequence and structural features[52]. Altogether, this scenario implies that oligonucleotides are unsuitable models for iM-forming sites in single-stranded genomic regions. This implication finds support in the observation of apparent differences in putative G-quadruplex structures and the $K^+$ dependence on G-quadruplex formation that appear to result when studying G-quadruplex motif in natural duplex, as opposed to the single-stranded form[53].

Secondly, the discrepancy may arise from the potential non-specificity and/or chaperoning activity of the iMab antibody. Apart from binding to iM states, iMab could bind to unfolded states in a sequence-specific manner or induce the formation of iMs (acting as a ligand) within the domain upon binding to the unfolded state. While these explanations have offered a straightforward way to account for the variance in the number of iM-forming sites observed using iMab compared to that observed using in-cell NMR, they lacked experimental support - until recently. At the time of submitting this revised manuscript, ref. 54 provided experimental evidence that the binding of iMab to DNA oligonucleotides is governed by the presence of runs of at

least two consecutive cytosines, irrespectively of the capacity of the sequence to adopt an iM structure.

Finally, the mentioned discrepancy can also be interpreted by considering the complementary (orthogonal) nature of in-cell NMR and iMab data. The detection method employed by iMab relies on chromatin fixation using paraformaldehyde, cell membrane permeabilization, and prolonged iMab staining at a temperature close to freezing[29]. Because paraformaldehyde cannot cross-link DNA secondary structures[55], and the near-freezing temperature is a critical factor leading to iM formation (cf. Supplementary Fig. 1), using i-Mab to detect iMs might become less effective when applied to cells from endothermic species, such as humans, which typically operate at considerably higher temperatures (37 °C). i-Mab could detect i-motifs induced by low temperatures and i-motifs that might naturally fold under physiological temperatures (37 °C), potentially leading to a significant overestimation of iM-forming sites in cells. In this context, the majority of the i-Mab-detected iMFPS would occur in an unfolded state in cells, and coil-to-iM/iM-to-coil transitions in the accessible regions of chromatin would be primarily under the control of the surrounding environment.

Several observations support this alternative scenario. For instance, there is evidence of proteins binding to unfolded states of iMFPS[42,56]. Furthermore, the environmental influence on iM formation persists even in the presence of mechanical forces that mimic those experienced by iMFPS in single-stranded chromatin; in a study employing a magnetic tweezers approach, Selvam et al. demonstrated that the impact of mechanical forces on iM stability is secondary to that of chemical factors such as pH, molecular crowding, and ionic strength[23]. These findings are also in line with observations using iMab; in a reference study, Zeerati et al.[29] observed that acidification of the intracellular environment led to an increased number of iMab-detected foci within chromatin. These observations suggest that chromatin's role in iM stability within the single-stranded regions of accessible chromatin is secondary to the relevant intracellular environmental conditions.

It is essential to emphasize that all the scenarios outlined regard that iM formation, accounting for coil-to-iM and iM-to-coil transitions in a chromatin context, is a consecutive (subordinate) step to competing processes related to strand separation (i.e., duplex-to-coil/coil-to-duplex transitions) and DNA-protein binding[17,57]. Evidence suggests that the chromatin microenvironment controls the duplex-to-coil/coil-to-duplex transitions in a double-stranded chromatin context; Selvam et al.[23] observed that decreasing superhelicity in torsionally constrained dsDNA fragments in an acidic solution led to duplex destabilization and an increased formation of iM, consistent with earlier findings by Hurley's group[22]. Later, the same research team demonstrated that the DNA duplex is also significantly destabilized under confined conditions marked by reduced water activity[24], aligning with prior observations[25]. However, these factors influence duplex-iM transition when the liberated strands (coils) tend to form sufficiently stable iMs under given environmental conditions[57].

Considering the aforementioned arguments and the complementary nature of experimental observations from both iMab and in-cell NMR, we propose a comprehensive model for iM formation within genomic iMFPS in human cells. In this model: (1) The number and locations of single-stranded accessible genomic sites prone to iM formation are dictated by cell-cycle-dependent chromatin states, including protein binding; (2) These sites are detected and identified using the iM-specific antibody (iMab) at near-freezing temperatures; (3) At the physiological temperature of 37 °C, the iM levels, representing the coil-to-iM transitions at the individual iMab-detected iMFPS, are primarily governed by cell-cycle-dependent properties (environmental factors) of the intracellular space. These factors push the equilibrium between the coil (unfolded) and iM (folded) states towards an unfolded state for the majority of iMFPS in the chromatin;

(4) The remaining iMFPS, which can form iM structures under physiological conditions in living human cells, can be identified using a simple criterion: $pH_T > 7$, where $pH_T$ corresponds to the pH transition point determined for oligonucleotide models of iMFPS in a reference buffer (25 mM KPO$_i$, 10.5 mM NaCl, 110 mM KCl, 1 mM MgCl$_2$, 130 nM CaCl$_2$) at 20 °C.

To conclude, our data and the proposed model support the idea of cell cycle-dependent iM formation in living human cells at physiological temperatures. However, they also indicate that the observed cell cycle dependency is of a dual origin, and iM formation concerns only a minuscule fraction of genomic sites displaying iM formation propensity. Regarding the dual origin of iM cell cycle dependency, our model implies that cell-cycle-dependent chromatin states control the number and location of sites with iM formation propensity, while changes in the properties of the intracellular environment during cell cycle progression regulate iM levels at these sites. Altogether, the proposed model indicates that most intragenic iMFPS in accessible single-stranded chromatin context are unfolded in living cells at physiological temperature, suggesting that the potential biological roles of many iMFPS are tied to their unfolded states. Nevertheless, establishing a causal link between the control of iM levels in cells and human physiology remains a significant challenge for future research. In terms of methodology, our study demonstrates for the first time that the original in-cell NMR approach, previously limited to monitoring structural equilibria in asynchronous suspensions of cells[34,58], can be directly applied to cells in defined physiological states.

## Methods

### DNA oligonucleotides

The DNA oligonucleotides employed in the present study are listed in Supplementary Table 1. The non-modified DNA oligonucleotides and their fluorescently (FAM of Cy3) 5′-labeled analogs were purchased from Sigma-Aldrich (USA) or Generi Biotech (Czech Republic). The FAM/Cy3-labeled oligonucleotides were dissolved in water to form 100 μM stock solutions and heated at 37 °C for 10 min. Oligonucleotides forming the dsDNA constructs were mixed in the ratio of 1:1.1 (hybrid-ds/iM FW: ds-DNA RV). To form the 5′-FAM-labeled ds DNA, the reverse strands were mixed with the FAM-labeled forward analogs in the abovementioned ratios. The samples were then annealed (5 min at 95 °C for non-labeled and 10 min at 37 °C for fluorescently labeled ds DNA oligonucleotides).

### Cell cultivation and synchronization

HeLa cells (Sigma-Aldrich, USA; Cat no 93021013) were cultured in Dulbecco's modified Eagle's medium (DMEM) (Sigma-Aldrich, USA) supplemented with 10% heat-inactivated fetal bovine serum (FBS) (HyClone, GE Life Sciences) and penicillin-streptomycin solution (P/S) (100 units penicillin and 0.10 mg streptomycin/mL) (Sigma-Aldrich, USA) under a 5% CO$_2$ atmosphere at 37 °C. After achieving ~80–90% confluency, the cells were passaged by washing with Dulbecco's phosphate-buffered saline (DPBS) (Sigma-Aldrich, USA) and by harvesting with 0.05% trypsin and 0.02% EDTA (Sigma-Aldrich, USA) in 1 x DPBS. The synchronization procedures were optimized and adjusted from Whitfield et al.[59] and Jackman & O'Connor[60] as follows: for synchronization at G1/S boundary (early S phase), cells were plated on 15-cm culture dishes to form ~25–30% confluency. When the cells adhered on the surface of the dishes (after approximately 4–6 h), they were washed with 1 x DPBS, followed by the application of new standard DMEM (with 10% FBS and P/S), containing 2 mM Thymidine (Sigma-Aldrich, USA) – for 18 h. After the first Thymidine block, the cells were released for 9 h by washing them twice in 1 x DPBS and leaving them in the standard DMEM. The second Thymidine block was applied by washing the cells with 1 x DPBS and adding the standard DMEM containing 2 mM Thymidine for 17 h.

For synchronization at the G2/M boundary (early M phase), cells were plated on 15-cm culture dishes to form ~ 50–60% confluency. When the cells adhered on the surface of the dishes (after approximately 4–6 h), they were washed with 1 x DPBS, followed by the application of new standard DMEM (with 10% FBS and P/S), containing Nocodazole reagent (Sigma-Aldrich, USA) at 0.4 µg/mL concentration – for 17 h.

## Preparation of in-cell NMR samples

Approximately $1 \times 10^8$ cells were used to prepare each in-cell NMR sample. The number of the 15-cm dishes used for cultivation differed according to each synchronization protocol (based on the final dish confluency on the sample's processing day). For early S cells, the number of dishes sufficient to obtain the required number of cells at the end of the synchronization procedure was 24. The number of M-phase synchronized dishes was 40, whereas, for the asynchronous (i.e., non-synchronized) cell population, it was only 12 dishes. The early S and asynchronous cells were washed in 1 x DPBS and trypsinized, whereas the mitotic cells were harvested by so-called "mitotic shake-off." All harvested cells were centrifuged in 50 mL falcon tubes at 220 g for 5 min. After centrifugation, the pellets in each falcon tube were pooled together by resuspending them in 1 x DPBS, forming one homogenous cell cycle-phase-specific sample. Samples were then counted in a Burker's chamber. Approximately $8 \times 10^6$ cells were placed in a 15 mL falcon tube, centrifuged at 220 g for 5 min and fixed in EtOH (see *Cell fixation in EtOH*). Simultaneously, the proper number of cells for transfection (~$1 \times 10^8$) was centrifuged again at 220 g for 5 min. The DNA oligonucleotides were introduced into the HeLa cells by electroporation using a BTX-ECM 830 system (Harvard Apparatus, USA). The pelleted cells were resuspended in 2.4 mL of electroporation buffer (EB buffer) (140 mM sodium phosphate, 5 mM KCl, 10 mM MgCl$_2$, pH = 7.0) containing 300 or 400 µM DNA and 10 µM FAM-labeled DNA. The cell suspension was divided into six 4-mm electroporation cuvettes (Cell Projects, UK). All samples were incubated on ice for 5 min before electroporation. Electroporation was conducted using two square-wave pulses (100 µs/1000 V; 30 ms/350 V) separated by a 5 s interval to achieve maximum transfection efficiency. After electroporation, the cells were incubated for 2 min at room temperature (RT), transferred into Leibovitz L15 −/− medium (no FBS/no antibiotics) (Sigma-Aldrich, USA) and centrifuged at 220 g for 5 min. The cells were resuspended in 10 mL of L15 −/− medium. A small portion of the cell suspension (~$6 \times 10^5$ cells) was used for flow cytometric (FCM) analysis and confocal microscopy analysis (see below) to evaluate the cell viability, transfection efficiency, and DNA localization. The rest of the cell suspension was centrifuged at 220 g for 5 min. After removal of the supernatant, the resulting cell pellet was resuspended in 550 µL of Leibovitz L15 −/− medium containing 10% D$_2$O and transferred into a Shigemi NMR tube (Shigemi Co., Tokyo, Japan). Before NMR analysis, the cells in the NMR tube were manually centrifuged using a "hand centrifuge" (CortecNet, France) to form a fluffy pellet at the bottom of the NMR tube.

## Bioreactor in-cell NMR

The pellet of cells after electroporation was mixed with 2% (w/w) SeaPrep agarose (Lonza, Switzerland) in DMEM at 37 °C, in v-v ratio 1:1. A HPLC capillary of 0.75 mm inner diameter was filled with the suspension and incubated on ice for 10 min. Upon gelling, a thread was pushed from the capillary into a 5 mm NMR tube filled with degassed bioreactor media: DMEM without NaHCO$_3$ (Sigma-Aldrich, USA) + 10% D$_2$O (Eurisotop, France) + 70 mM HEPES (Sigma-Aldrich, USA) + 1x ZellShield (Minerva Biolabs, Germany). Medium flow through the sample was set to 50 µl/min, using an HPLC pump-driven system that draws the medium from a reservoir heated in a water bath through a vacuum-degassing chamber and brings the fresh medium to the bottom of the NMR tube through a central glass capillary[61].

## Flow cytometry

For FCM analysis, ~$10^5$ cells were resuspended in 200 µL of DPBS buffer (Sigma-Aldrich, USA). To distinguish the apoptotic, dead cells or cells with compromised membrane integrity from the living cells, the sample was stained with 1 µL (1 mg/mL) of propidium iodide (PI) (Exbio, Czech Republic). The total amount of $10^4$ cells was analyzed by a BD FACSVerse flow cytometer using BD FACSuite software V1.0.6 (BD Biosciences, USA). To detect the fluorescently (FAM) labeled DNA and thus evaluate the transfection efficiency, the excitation and emission wavelengths were 488 nm and 527/32 nm, respectively. PI was excited at 488 nm to evaluate the cell viability, and the emission was detected at 700/54 nm. For analyzing fixed cells stained with PI and MPM-2, the excitation wavelength was set at 488 nm for PI to visualize the DNA content and 640 nm for MPM-2 to quantify the number of mitotic cells in the sample. The PI and MPM-2 emissions were detected at 700/54 and 660/10 nm, respectively. Supplementary Figs. 12 and 13 exemplify the employed gating strategy.

## Confocal microscopy

For confocal microscopy, ~$5 \times 10^5$ cells were placed onto a 35-mm glass-bottomed dish (ibidi GmbH, Germany) pre-coated with 0.01% poly-L-lysine (Sigma-Aldrich, USA). The cell drop was then immersed in 2 mL Leibovitz L15 −/− medium containing 1 µL/mL Hoechst dye (Sigma-Aldrich) to visualize the cell nuclei. All microscopy images were obtained using a Zeiss LSM 800 confocal microscope with a 63x/1.2 C-Apochromat objective. Images were taken in transmission mode with an excitation wavelength of 488 nm, and the fluorescently (FAM) labeled DNA was detected at emission wavelengths of 480–700 nm. The excitation wavelength for Hoechst was set to 405 nm, and the emission was detected at 400–480 nm. The images were collected and analyzed in ZEN Blue 2.6 and 3.1 software.

## Cell fixation in EtOH

*Before* fixation, cells were first washed with 1 x DPBS and centrifuged at 220 g for 5 min. The pellet was thoroughly resuspended in 300 µL 1 x DPBS and finally, 2.5 mL of 70% ethanol (EtOH) was added. The sample was then vortexed and stored at −20 °C for up to 1 week but at least 1 h prior to staining.

## PI and MPM-2 staining

Fixed cells were washed twice in ice-cold 1 x DPBS (5 mL and 1 mL), with the centrifugation steps at 17,000 g and 4 °C for 5 min. After the second centrifugation, the sample was resuspended in 100 µL 1 x DPBS and incubated for 1 h in the dark with ~3 µL of the MPM-2 antibody (anti-phospho-Ser/Thr-Pro, Cy5 conjugate) (Millipore, Sigma-Aldrich, USA; Cat no 16-220, Lot no 3134673) that stains cells containing mitosis-specific phophorylations. Next, the sample was washed twice in 1 mL of ice-cold 1 x DPBS, centrifuged at 17,000 g, 4 °C for 5 min, and resuspended in 200 µL of 1 x DPBS. 10 µL of propidium iodide (PI) was added to the sample to visualize the DNA content, and 0.5 µL of RNAse A (10 mg/mL) (Thermo Fischer Scientific, USA) to remove RNA. The sample was incubated at 37 °C shaking in the dark for approximately 40 min and subsequently taken for FCM analysis.

## NMR spectroscopy

The NMR spectra were measured at 600, 850, or 950 MHz using Bruker Avance NEO spectrometers (Bruker, Corporation, Billerica, USA) equipped with a quadruple (600 MHz) or a triple-resonance (850 and 950 MHz) inverse cryogenic probe. In-vitro 1D $^1$H spectra accompanying the in-cell data were acquired in an electroporation buffer (140 mM sodium phosphate, 5 mM KCl, 10 mM MgCl$_2$, pH = 7.0) with 10% D$_2$O. In-vitro 1D $^1$H NMR spectra mapping the pH or temperature-

**Table 2 | The list of primary and secondary antibodies used in Shift-Westen-Blot assay**

| Target | Origin | Conjugated | Dilution | Manufacturer | Cat. number |
|---|---|---|---|---|---|
| Primary antibodies | | | | | |
| PCBP2 | rabbit polyclonal | | 1:1000 | Origene | TA308051 |
| hnRNP K | mouse monoclonal | | 1:1000 | Abcam | ab39975 |
| hnRNP A1 | mouse monoclonal | | 1:1000 | Sigma Aldrich | R4528 |
| Secondary antibodies | | | | | |
| rabbit IgG | goat | HRP | 1:10000 | Jackson ImmunoResearch | 111-035-003 |
| mouse IgG | goat | HRP | 1:10000 | Jackson ImmunoResearch | 115-035-003 |

*HRP* Horseradish Peroxidase.

dependent behavior were acquired in an intracellular (ICB) buffer (25 mM potassium phosphate, 10.5 mM NaCl, 110 mM KCl, 1 mM MgCl$_2$, and 130 nM CaCl$_2$) with 10% D$_2$O. In-cell NMR spectra were acquired in Leibovitz L15 −/− medium containing 10% D$_2$O. The DNA concentrations in the in-vitro samples were 100 μM, unless stated otherwise. (in-cell) 1D $^1$H NMR spectra were acquired using a 1D $^1$H JR-echo (1-1 echo) pulse sequence[62], with zero excitation set to the resonance of water and the excitation maximum set to 15.3 ppm. The total experimental time to acquire in-cell $^1$H NMR spectra was ~10 min, unless stated otherwise. All NMR spectra were baseline-corrected and processed with the exponential apodization function. The NMR spectra were acquired using Topspin 4.0.6 and processed using MestReNova v14.0.1 (Mestrelab Research, Spain). After the acquisition of the in-cell NMR spectra, the 1D $^1$H NMR spectra of the supernatant in each NMR tube were measured (using the same NMR parameters as were used to acquire the in-cell NMR spectrum) to assess possible leakage of the transfected DNA from the cells. At the end of the experiment, the cells were taken from the NMR tube and subjected to FCM analysis to assess cell mortality during the measurement. A small fraction of cells was also fixed in EtOH again and subsequently stained with PI and MPM2 (see above) to control whether the cells were already escaping from the synchronized populations.

## CD spectroscopy

Circular dichroism experiments were performed on a JASCO J-815 spectropolarimeter equipped with a JASCO PTC-423S temperature controller using a 0.1 cm path-length quartz cuvette. Briefly, a 50 μM oligonucleotide solution was prepared for each sample in ICB buffer (25 mM potassium phosphate, 10.5 mM NaCl, 110 mM KCl, 1 mM MgCl$_2$, 130 nM CaCl$_2$) at appropriate pH. Samples were heated at 95 °C for 5 min and let cool down overnight. pH titrations were performed by adding increasing concentrations of HCl to the samples and equilibrating them for 5 min after each addition. Every titration step was followed by measuring the CD spectrum of the DNA solution. CD spectra were recorded at 20 °C in a wavelength range of 220–320 nm using the following parameters: scanning speed of 100 nm/min; bandwidth of 2 nm; data interval of 0.5 nm; response of 1 s. The buffer contribution was subtracted from each CD spectrum after the acquisition. The acquired dichroic signal was finally converted in molar ellipticity ($[\Theta] = \deg \times cm^2 \times dmol^{-1}$) calculated using the DNA concentration in solution and adjusted according to the dilution factor. CD melting experiments were performed on the same instrument using a 1 cm pathlength quartz cuvette. 5 μM oligonucleotide solutions were prepared in ICB-cacodylate buffer (25 mM sodium cacodylate, 10.5 mM NaCl, 110 mM KCl, 1 mM MgCl$_2$, 130 nM CaCl$_2$) at different pH (specifically, at pH 7, 6.5, 6, 5.5, 5), heated at 95 °C and then cooled-down overnight. Each sample was equilibrated at the measurement starting temperature for 20 min before the acquisition. Samples were measured using the following parameters: temperature ramp of 0.3 °C/min; data pitch every 0.5 nm; bandwidth of 2 nm. The temperature range was varied according to the stability of the folding under examination. The denaturation process was

monitored at 288 nm. Raw experimental data were processed using JASCO Spectra Manager 2.09.10 software and then analyzed using GraphPad Prism 8.

## Preparation of samples for EMSA

HeLa cells were cultured in Dulbecco's modified Eagle's medium (DMEM) (Sigma-Aldrich, USA) supplemented with 10% heat-inactivated fetal bovine serum (FBS) (HyClone, GE Life Sciences) and penicillin-streptomycin solution (P/S) (100 units penicillin and 0.10 mg streptomycin/mL) (Sigma-Aldrich, USA) under a 5% CO$_2$ atmosphere at 37 °C. After achieving ~90% confluency, the cells were harvested, counted on a Burker's chamber, and ~1.6 × 10$^7$ cells (~120 μL of cell pellet) were electroporated with 5 or 50 μM (unless indicated otherwise) DNA oligonucleotides labeled with Cy3 dye at their 5' ends (purchased from Generi Biotech, Czech Republic), according to the procedure described above. After the electroporation, samples were washed twice in 1 mL Leibovitz L15 −/− medium (no FBS/no antibiotics) (Sigma-Aldrich, USA) and centrifuged at 220 g for 5 min. After centrifugation, the supernatant was discarded, and the pellets were lysed as follows: the pellets were resuspended in 2 package cell volume of buffer A (10 mM HEPES, pH 7.9, 1.5 mM MgCl$_2$, 10 mM KCl, 300 mM Sucrose, 0.5% NP-40) with cOmplete, EDTA-free Protease Inhibitor Coctail Tablet (Sigma-Aldrich, USA) and left on ice for 10 min. 2/3 package cell volume of buffer B (20 mM HEPES, pH 7.9, 1.5 mM MgCl$_2$, 420 mM NaCl, 0.2 mM EDTA, 2.5% glycerol) with cOmplete, EDTA-free Protease Inhibitor Coctail Tablet (Sigma-Aldrich, USA) was added, and the mixture was then sonicated with a microtip for 5 s at amplitude 95%, and subsequently centrifuged at 8000 g for 5 min. The collected supernatant was then stored at −20 °C.

## The gel electrophoresis mobility shift assay (EMSA) in agarose gel

For EMSA, samples were mixed with 6 x Orange G DNA Loading Dye (Sigma-Aldrich, USA) and run on a 0.8% agarose gel at 75 V for ~50 min, in 1 x TBE buffer (89 mM Tris, 89 mM boric acid, 2 mM EDTA, pH 8.0). The resulting gels were then scanned by a GE Typhoon™ FLA 9500 Fluorescent Image Analyzer, and the image contrast adjustments were performed in Adobe Photoshop 2020. The uncropped and unprocessed scans of the gel are shown in the Source Data of the article or Supplementary Figs. 14, 15, and 16.

## Identification of proteins interacting with DNA oligonucleotides

A protocol modified from ref. 41 was used for electrophoretic separation and transfer of protein-DNA complexes to a membrane. In brief, lysates from cells electroporated with Cy-3 labeled oligonucleotides were loaded on 6% polyacrylamide gel (acrylamide: bisacrylamide ratio 29:1) in 1x TBE and separated by electrophoresis in 1x TBE running buffer (89 mM Tris, 89 mM boric acid, 2 mM EDTA). Migration of the DNA in the gel was visualized via the fluorescent Cy-3 label (ChemiDoc™ MP Imaging System, BIO-RAD, USA). Protein-DNA complexes were transferred from the gel to the PVDF membrane

(Immun-Blot® PVDF Membrane, BIO-RAD, USA) using semi-dry blotting in 1x transfer buffer (48 mM Tris, 39 mM glycine, 20% methanol, pH 8.5) at a constant current $0.8\,mA/cm^2$ for 90 min. (Trans-Blot® Turbo™ Transfer System, BIO-RAD, USA). Upon membrane blocking with 5% skim milk and 5% BSA in TBST (150 mM NaCl, 50 mM TRIS-HCl, pH 7.8, 1 M EDTA, 0.1% Tween-20), proteins on the membrane were detected by immunostaining with the primary antibody in the blocking solution, followed by washing in TBST and incubation with horseradish peroxidase-conjugated secondary antibody in the blocking solution. After washing with TBST, protein signals were visualized by incubation of the membrane in ECL substrate (Clarity™ Western ECL Substrate, BIO-RAD, USA), and chemiluminescence was detected with Odyssey® Fc Imaging System (Li-COR, USA). Upon image acquisition, brightness and contrast were adjusted using Image Studio 3.1.4 software. Lists of primary and secondary antibodies are given in Table 2. The uncropped and unprocessed scans of the gel are shown in the Source Data of the article or Supplementary Figs. 14, 15, and 16.

## Data availability
All data generated in this study are available within the Article and Supplementary Information. The raw NMR data generated in this study have been deposited in the public data repository[63]. The primary CD data generated in this study are provided as a Source Data file. Source data are provided with this paper.

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

## Acknowledgements

This project was supported by grants from the Czech Science Foundation (GX19–26041X to L.T.), European Regional Development Fund, with financial contribution from MEYS CR (SYMBIT: CZ.02.1.01/0.0/0.0/15_003/0000477 to S.F.T. and J.L.M., MSCAfellow2@MUNI to R.R. [CZ.02.2.69/0.0/0.0/20_079/0017045]), European Union—Next Generation EU (project National Institute for Cancer Research; Program EXCELES, ID Project No. LX22NPO5102 to S.F.T.), ANR grant [ANR-21-CE44-0005-01] 'ICARE' to J.L.M., and the National Science and Engineering Research Council of Canada (Discovery Grant to M.J.D.). The authors also acknowledge the institutional projects enabling access to research infrastructure: APPID (815) financed by Instruct, iNEXT-Discovery [871037] financed by Horizon 2020 program of the European Commission, Josef Dadok National NMR Center of CIISB, Instruct-CZ Center, supported by MEYS CR (LM2023042) and European Regional Development Fund-Project „UP CIISB" (No. CZ.02.1.01/0.0/0.0/18_046/0015974), core facility CELLIM supported by MEYS CR (LM2023050 Czech-BioImaging), and the Flow Cytometry Laboratory at CEITEC MU supported by the EATRIS-ERIC-CZ research infrastructure (LM2023053 funded by MEYS CR). *Disclaimer*: the iNEXT-Discovery project has received funding from the European Union's Horizon 2020 research and innovation program under grant agreement No 871037. This publication reflects only the author's view, and the Research Executive Agency/ the European Commission is not responsible for any use that may be made of the information it contains.

## Author contributions

S.F.-T., J.-L.M. and L.T. conceived and designed the study; P.V., R.E.-K., M.J.D., M.L.Z., I. S.-C., and C.G. contributed to in vitro sample preparation, and synthesis of modified oligonucleotides used in initial NMR-based screenings; P.V., J.R., and S.-F.T. performed EMSA and WB-based

experiments; P.V., E.I., J.R., S.D., T.L., and S.F.-T. performed in-cell NMR experiments; R.R. performed CD experiments; P.V., E.I., S.D., R.R., T.L., and S.F.-T., performed data processing; L.T., S.F-T., J.-L.M., M.J.D., and C.G. supervised research activities; P.V., E.I., R.R., T.L., and S.F.-T. took care of data visualization; L.T. and S.F.-T. wrote the first original draft; all authors contributed to the manuscript review & editing; L.T., R.R., M.J.D., and J.-L.M. secured the research funding.

## Competing interests

The authors declare no competing interests.
