## [Peer Review File · Nature Communications]

REVIEWER COMMENTS

Reviewer #1 (Remarks to the Author):

A method to assess the effect of the cellular environment on i-motif (iM) formation using in-cell NMR is presented. Oligonucleotides with the iM sequence directly linked to a double-stranded (ds) DNA segment are transfected into asynchronous and synchronized cells. The data show that iM sequences that do not fold in vitro at physiological pH are not folded in cells, whereas those with high transition pH levels can be observed in cells. It is shown that iMs can be refolded in the cell under the right conditions (e.g., temperature < 37°C). Finally, interaction with cellular proteins previously shown to bind C-rich DNA is reported.

Comments:

The method primarily assesses the folding state of exogenous iM-forming sequences of interest. The dynamicity of non-B DNA structures suggests that their folding and unfolding is highly dependent on the binding to proteins during cellular processes, which are fundamental to assess the biological relevance of iMs (DOI: 10.1021/ja410934b). The model presented here, therefore, cannot be considered comprehensive of the many variables that are involved in iM formation in the genome. It is also not clear how the cited low temperature incubation of the iMab antibody used in cells to show the presence of iMs, both by immunofluorescence in fixed cells and the new technique CUT&TAG in live cells would induce iMs from a chromatin-embedded double-stranded DNA. Is there evidence that this could happen?

In any case, this manuscript adds new information to the still rather underdeveloped field of iMs.

Issues to be addressed.

The construct tested is formed by a duplex region linked to the iM-forming sequence. The dsDNA fragment is reported to be "environmentally insensitive". How was this proven? No controls with the ds part linked to unstructured DNA or other non-B DNA structures (e.g. G-quadruplexes) are shown. Has this kind of construct ever been used in in-cell NMR?

How come that the ds DNA does not interact with proteins? I would expect DNA repair proteins to be found. This is quite troubling since it may indicate that proteins do not bind to the construct, not even to the iM part. Since proteins are expected to be involved in iM folding, this could show that a bias is present in the method.

Are there controls to show that proteins bound to DNA within the chromatin context can be displaced by the presence of the synthetic construct?

Is it possible that the quantity of the proteins available in the cell binds only a fraction of the construct? In such a case I would expect the NMR signal to come mostly from the unbound construct.

Are there controls to address this issue?

To assess the binding to the cellular lysate, three known C-rich binding proteins were investigated, hnRNPK, hnRNPA1 and PCPB2. Proteins that were proven not to bind C-rich DNA (e.g., Nucleolin) need to be tested as negative controls.

It would be interesting to test a longer (CCCTAA)_n sequence, as this is present in many repeats at the telomeres.

Regarding Figure S6, also shown as A1 in the response to reviewers file, it is not clear why the samples containing the oligo alone do not migrate into the gel and why the samples do not show bands corresponding to the free oligo (unbound). This internal control is necessary to show that the proteins only are blotted onto the PVDF membrane.

As for the synchronized cells, in figure S7-IIB the blue staining shows the DNA not the nucleus, especially in the M phase where the nucleus is not present, due to the ongoing cell division. Also discuss that the protein content is expected to largely vary among cell phases.

The title is misleading, as it leads the reader to think it refers to iMs present in the cellular genome. It should be revised highlighting the fact that the experiments are performed on synthetic constructs in the cell environment.

Response to referee's #1 general comments:

The method primarily assesses the folding state of exogenous iM-forming sequences of interest. The dynamicity of non-B DNA structures suggests that their folding and unfolding is highly dependent on the binding to proteins during cellular processes, which are fundamental to assessing the biological relevance of iMs (DOI: 10.1021/ja410934b). The model presented here, therefore, cannot be considered comprehensive of the many variables that are involved in iM formation in the genome.

The reviewer's comment is valid. Like other existing technologies, including iMab, the in-cell NMR approach cannot comprehensively address i-motif genesis within cells. However, it offers unique complementary insights that remain unattainable through other methods. Consequently, not only the strengths but also the limitations of the in-cell NMR approach have been deliberated in the manuscript.

It's worth noting that the main conclusions concerning i-motif-protein binding from the study indicated by the reviewer (DOI: 10.1021/ja410934b) were later proved to be incorrect (DOI: 10.1002/cbic.201700390).

It is also not clear how the cited low-temperature incubation of the iMab antibody used in cells to show the presence of iMs, both by immunofluorescence in fixed cells and the new technique CUT&TAG in live cells, would induce iMs from a chromatin-embedded double-stranded DNA. Is there evidence that this could happen? In any case, this manuscript adds new information to the still rather underdeveloped field of iMs.

To address this comment, clarification is necessary as it contains inaccuracies: Neither the in-cell NMR nor iMab data (on fixed/unfixed cells) pertain to double-stranded DNA. These methodologies focus on accessible single-stranded (ss)DNA chromatin regions with a propensity for i-motif formation (in in-cell NMR experiments, these regions are approximated using single-stranded oligonucleotides). The temperature-dependent nature of i-motif formation in ssDNA context under diluted in vitro conditions has been recognized for nearly 30 years. We have demonstrated that the temperature dependence of i-motif formation in ssDNA context in cells closely mirrors that observed in vitro. While it has been an implicit presumption in interpreting iMab data that i-motif stability in the context of chromatin is significantly increased (for an in vitro equivalent of at least 25 °C) compared to in vitro situation, none of the iMab-based studies have ever validated this assumption. Nevertheless, the absence of unbiased technology for such measurements could account for this limitation.

We admit that our original text has not addressed the unique position of the CUT & TAG approach using iMab with respect to the in-cell NMR approach. Please see our response to reviewer #1 comment #3A addressing this issue.

Note to the reviewer: The comment is biased, taking iMab specificity to iMs granted and iMab-based observations as reference. However, roughly two weeks ago (November 21st, 2023), a new set of experimental data concerning iMab was released, revealing that, contrary to the initial claims regarding iMab's i-motif specificity, this antibody actually recognizes and binds to C-rich DNA sequences, regardless of their capacities to form iM (Boissieras et al. iMab Antibody Binds Single-Stranded Cytosine-Rich Sequences and Unfolds DNA i-Motifs. DOI: <https://doi.org/10.1101/2023.11.21.568054>). These findings align with our in-cell NMR observations and support an alternative understanding of iMab data. This perspective integrates the detection of accessible genomic sites in the strand-separated state (iMab data), with the

propensity for these sites to form i-motif under physiological temperatures within the complex intracellular environment (in-cell NMR data). *These combined observations are the sole supporting evidence of i-motif formation in cells, emphasizing the necessity of considering both iMab and in-cell NMR data rather than relying on either dataset in isolation.*

Issues to be addressed according to reviewer #1.

1A] The construct tested is formed by a duplex region linked to the iM-forming sequence. The dsDNA fragment is reported to be “environmentally insensitive.”

In the reviewer’s comment, there is a criticism of the inappropriate use of the term “environmentally insensitive,” which might give the impression to readers that the dsDNA fragment is insensitive to all possible environmental conditions and sequence/structure contexts. We apologize for the confusion. In our manuscript's context, the dsDNA fragment's insensitivity is expressed relative to the structure of the i-motif forming segment and range of close-to-physiological environmental conditions spanning pH (6.0 - 7.5) and temperature (4-37 °C) and accounting (via in-cell NMR readout) for other native non-specific environmental factors such as a native molecular crowding or osmotic stress due to solutes.

1B] How was this proven? No controls with the ds part linked to unstructured DNA or other non-B DNA structures (e.g. G-quadruplexes) are shown.

Controls for the relative insensitivity of dsDNA to pH perturbations compared to the i-motif forming segment (folded/unstructured) have been presented in Figures 1D, 1E, 2A, and S3 presenting pH-dependent in vitro NMR and CD spectra of hybrid-ds/iM constructs, respectively. Please see Figure 1E pertinent to hybrid-ds/hT121-6, for example: while the i-motif specific signal ($\delta \sim 15.4$ ppm) is reduced for $\sim 95\%$ in response to the pH change from 6.22 to 7.46, the imino signal intensities corresponding to the dsDNA segment ($\delta \sim 12.0-14.2$ ppm) are essentially unaltered. These controls have been discussed in detail in the original text but were probably overlooked by the reviewer. Similarly, the relative insensitivity of dsDNA compared to temperature perturbations when compared to the i-motif forming segment (folded/unstructured) can be directly perceived from in-cell and in vitro NMR data of hybrid-ds/iM constructs presented in Fig. 1G, H, 2B, and Fig. S2, respectively. For instance, referring to Fig. 1H, the iM-specific NMR signals (highlighted in the red box) observed at 20°C are essentially reduced to zero when the temperature is increased to 37°C. Conversely, the alterations in the NMR signal intensities (integral intensities – considering temperature-dependent line broadening and the inherently low signal-to-noise ratio in the in-cell NMR spectra) corresponding to the dsDNA segment (gray box) are minimal.

Action taken regarding the revised manuscript: To prevent further misunderstanding, we modified the text to explicitly state that the “environmental insensitivity” of the dsDNA segment is expressed relative to the iM structure and range of defined environmental conditions. To further support our statements on “environmental insensitivity” (relative to the iM), we included in vitro NMR spectra of hybrid-ds/hT121-6 acquired in the intracellular buffer (ICB) and crude cellular homogenate at low (iM segment structured) and elevated temperatures (iM segment unfolded) into Supporting Information and referenced these new data in the text (new Fig. S4). These data confirmed the ds-segment's minimal sensitivity to temperature changes between 4 to 37 °C, in contrast to the iM probe (hT121-6), in both in vitro

and cell-like settings. Additionally, they support the ds-segment's suitability to normalize the i-motif/single-strand equilibrium within cells.

2] Has this kind of construct ever been used in in-cell NMR?

Yes, a construct of identical design (hybrid-ds/iM) was recently employed as a probe to evaluate a novel polymeric gel matrix, specifically poly(D, L-lactide)-b-poly(ethylene glycol)-b-poly(D,L-lactide) (PLA-PEG-PLA), intended for utilization within in-cell NMR bioreactors (DOI: 10.1007/s10858-023-00422-7).

3A] How come that the ds DNA does not interact with proteins? I would expect DNA repair proteins to be found. This is quite troubling since it may indicate that proteins do not bind to the construct, not even to the iM part. Since proteins are expected to be involved in iM folding, this could show that a bias is present in the method. Are there controls to show that proteins bound to DNA within the chromatin context can be displaced by the presence of the synthetic construct? Is it possible that the quantity of the proteins available in the cell binds only a fraction of the construct? In such a case I would expect the NMR signal to come mostly from the unbound construct.

To a large extent, the reviewer's comment is self-addressed. The in-cell NMR signals indeed arise exclusively from the unbound DNA—a point we stressed in the original text: "...NMR signals report on unbound DNA." This principle is foundational in nucleic acid in-cell NMR analysis (DOI: 10.1002/1873-3468.13054; 10.1007/128_2012_332; 10.1002/anie.201311320; 10.2142/biophysico.BSJ-2020006; 10.1007/s12551-020-00664-x).

The possibility to isolate signals from unbound DNA in in-cell NMR spectra is vital for direct comparison with iMab-based data and central to our work. While the in-cell NMR approach isolates the information on unbound DNA via the dependence of the NMR signal on molecular correlation time (molecular tumbling/size), in the iMab experiment, the unbound i-motif forming regions in chromatin are "separated" from the i-motif forming regions bound by proteins via utilization of chemical cross-linking; the chemical cross-linking allows iMab to specifically detect iMs in single-stranded accessible (protein-unbound) chromatin regions.

To conclude, like in iMab studies following the original protocol by Zeerati et al (DOI: 10.1038/s41557-018-0046-3), the protein-bound fraction isn't within the scope of our present study.

However, we suspect the reviewer's comments, in fact, aimed to highlight the existence of two fundamentally different procedures for using iMab, a distinction we overlooked in our original text. While most iMab studies have involved the cross-linking step, a recent study by Zanin et al. (DOI: 10.1093/nar/gkad626, September 2023) employed a modified protocol using iMab without chemical cross-linking. The iMab data derived with and without cross-linking are not comparable directly.

Action taken regarding the revised manuscript: We apologize to the reviewer for not commenting. To account for the distinction between in-cell NMR/iMab-cross-linking-based data and recent iMab data by Zanin et al. (DOI: 10.1093/nar/gkad626), we modified the text of the revised MS as follows: However, it's important to note that this interpretation is confined to the original iMab protocol that involves a cross-linking step.³⁰ Employing the modified iMab protocol introduced by Zanin et al.,³³ which excludes the cross-linking, iMab can also identify iMFPS in the single-stranded chromatin regions bound by proteins; iMab might outcompete cell-cycle specific proteins that bind to these iMFPS."

3B] Are there controls to address this issue?

The evidence that the in-cell NMR readout comes from unbound (free) DNA is in main text Figures 1 and 2, it particularly follows from comparing the corresponding in vitro and in-cell NMR spectra acquired at the same temperature. It mainly concerns two fundamental NMR observables: isotropic chemical shielding defining the positions of the NMR signals in the spectrum, and transversal relaxation time defining NMR signal linewidth.

A] Signal positions: Unbound (free) and protein-bound DNA, from the point of view of their electronic structure, are two different entities. NMR spectrum as an electronic structure reporter can be regarded as a unique atomically resolved fingerprint of the structure; this constitutes the basis for using NMR spectroscopy in structure analysis. Fingerprints matching: Correspondence of the in vitro NMR spectra patterns of DNA constructs acquired IC buffer (in the absence of the proteins) with the corresponding in-cell NMR spectra evidence detection of the single species, free DNA (cf. Figures 1 and 2).

B] NMR signal linewidth: For observation of DNA construct (any biomolecule) using solution-state in-cell NMR spectroscopy, the DNA of interest has to tumble sufficiently fast in the intracellular environment. However, interaction with other large cellular components, such as proteins, slows the tumbling rate, thus resulting in markedly broad or undetectable resonance lines. Figures 1 and 2 show that the line widths of in-cell NMR signals are less than 2.5 times broader than those acquired in buffer solution; the level of observed line broadening corresponds to the expected generic impact of the intracellular matrix, including viscosity, on the molecule's rotational diffusion (DOI: 10.1021/ja0112846; 10.1002/chem.201301657).

Action taken regarding the revised manuscript: Recognizing that not all readers are acquainted with (in-cell) NMR spectroscopy and understanding that the manuscript cannot replace NMR spectroscopy textbooks, we have made revisions to include references that delve into these fundamental aspects of interpreting in-cell NMR spectra. The text was modified as follows: “.....Matching signal patterns (signal positions) (10.1007/128_2012_332) and similar linewidth (10.1002/anie.201311320) among the corresponding signals in in-vitro and in-cell NMR spectra acquired at 20 °C (Fig. 1D, E, G, H) evidence that in-cell NMR readout reports on the intracellular space's free (unbound) hybrid-ds/iMs.

4] To assess the binding to the cellular lysate, three known C-rich binding proteins were investigated, hnRNPK, hnRNPA1 and PCPB2. Proteins that were proven not to bind C-rich DNA (e.g., Nucleolin) need to be tested as negative controls.

In the reviewer's comment, there's a call for a negative control in the native PAGE-based experiment, suggesting using a proven protein like Nucleolin that doesn't bind to C-rich DNA. While we acknowledge the importance of a negative control, it's essential to note that the **experiment in question already incorporates a negative control**, which the reviewer has overlooked.

There are at least two equivalent approaches to incorporate a negative control into the experiment: a) employing a protein that doesn't bind to C-rich DNA, as suggested by the reviewer, and b) using a DNA oligonucleotide that isn't recognized by C-rich binding proteins. In our experiment, we chose the latter option due to the absence of proteins definitively established as not binding to C-rich DNA. Nucleolin, proposed by the reviewer, well illustrates the situation; Nucleolin is considered a G-quadruplex binding protein, yet it also exhibits binding to C-rich (iMFPs) DNA (DOI: 10.1074/jbc.M109.018028), albeit with comparably lower affinity. This aspect renders Nucleolin unsuitable for negative control in our setup.

Our DNA-protein binding assay employed a G-rich (G-quadruplex-forming) oligonucleotide from the human c-Myc promoter as a negative control. Upon introducing the cellular lysate, the electrophoretic migration of the G-rich oligonucleotide was retarded (Fig. 2C LEFT panel, Fig. S7A and B). Subsequently, upon transfer to a membrane and analysis with specific antibodies, no complex formation (negative control) between the experimentally validated C-rich binding proteins and the G-rich oligonucleotide was observed (Fig. 2C RIGHT panel, Fig. S7C).

Action taken regarding the revised manuscript: None; the experiment already incorporates negative control.

5] It would be interesting to test a longer (CCCTAA)_n sequence, as this is present in many repeats at the telomeres.

We agree with the reviewer that numerous extended C-rich telomeric DNA aspects could merit further examination and investigation. However, these aspects are considerably beyond the scope of the current study. Our primary focus in this study is to complement and provide an orthogonal perspective to iMab, for which phage selections were conducted to isolate binders to the human telomere i-motif (hTelo i-motif), a telomeric (CCCTAA)_n sequence based on n=4. (DOI: 10.1038/s41557-018-0046-3).

Action taken regarding the revised manuscript: None.

6] A] Regarding Figure S6, also shown as A1 in the response to reviewers' file, B] it is unclear why the samples containing the oligo alone do not migrate into the gel and why the samples do not show bands corresponding to the free oligo (unbound). This internal control is necessary to show that the proteins are only blotted onto the PVDF membrane.

Addressing this comment requires clarification; the reviewer misread the Figure legend:

A] Figure A1, in response to the reviewers' file, constituted only part of the original Figure S6, specifically three out of four panels in S6B displaying blotted native PAGE to the PVDF membrane; blotted proteins were visualized using primary and secondary antibodies. It follows that **Figure A1 (and the corresponding panels in S6B) displayed only anti-body-recognized proteins.**

B] Unlike proteins, DNA oligos covalently linked to Cy3 were visualized using Cy3 fluorescence directly in the agarose gel (the original Figure S6A) and native PAGE (the original Figure S6B; first panel). As could be seen in the original Fig. S6A (agarose, Cy3 visualized, gel) and the first panel in S6B (Cy3-visualized native PAGE) - samples containing the oligo alone *did migrate into the gel*, and *they did show bands* corresponding to the free oligo (unbound).

Please note that the original Fig. S6 corresponds to Fig. S7 in the revised version of SI.

Action taken regarding the revised manuscript: To improve the presentation of electromigration data and prevent further confusion, we split panel B in the original Fig. S6. In the revised version (Fig. S7), panel B displays Cy3-visualized native PAGE (only DNA). In contrast, panel C displays blotted native PAGE to PVDF membrane stained with the protein-specific antibodies as indicated (only proteins).

7] As for the synchronized cells, in Figure S7-IIB, the blue staining shows the DNA, not the nucleus, especially in the M phase, where the nucleus is not present due to the ongoing cell division.

Action taken regarding the revised manuscript: We thank the reviewer for pointing out the use of an improper term. The respective text in the figure legend (Fig. S8 corresponding to the original Fig. S7) was corrected to indicate that the blue color marks Hoechst-stained genomic DNA.

8] *Also discuss that the protein content is expected to largely vary among cell phases.*

We believe that addressing the reviewer's comment #3 (as mentioned earlier) renders this suggestion unnecessary. As clarified earlier, the interpretation of in-cell NMR data and the primary focus of the manuscript do not involve DNA-protein binding. Therefore, discussions about cell cycle-dependent changes in the content of specific proteins, aside from their generic effect via molecular crowding, are irrelevant to the manuscript's content.

Action taken regarding the revised manuscript: None.

9] *The title is misleading, leading the reader to think it refers to iMs present in the cellular genome. It should be revised, highlighting that the experiments are performed on synthetic constructs in the cell environment.*

With all due respect to the reviewer's opinion, the authors would like to retain the title in its current form, as it already implicitly includes the notion of oligonucleotides as models of iMPFs through the phrase 'in-cell NMR insight.' Information gathered from (in-cell) NMR spectroscopy, much like other high-resolution methods such as X-ray and neutron diffraction, is inherently limited to oligonucleotide models. We highlight in the MS abstract that the experiments are performed on synthetic constructs in the cell environment.

Action taken regarding the revised manuscript: None.

REVIEWERS' COMMENTS

Reviewer #1 (Remarks to the Author):

The reviewer's main comments have been addressed